# α-Synuclein Aggregates in the Nigro-Striatal Dopaminergic Pathway Impair Fine Movement: Partial Reversal by the Adenosine A_2A_ Receptor Antagonist

**DOI:** 10.3390/ijms24021365

**Published:** 2023-01-10

**Authors:** Qionghui Cai, Na Xu, Yan He, Jiamin Zhu, Fenfen Ye, Zhi Luo, Ruojun Lu, Linshan Huang, Feiyang Zhang, Jiang-Fan Chen, Yan Li

**Affiliations:** 1Department of Neurology, The Second Affiliated Hospital and Yuying Children’s Hospital of Wenzhou Medical University, Wenzhou 325027, China; 2The Molecular Neuropharmacology Laboratory and the Eye-Brain Research Center, The State Key Laboratory of Ophthalmology, Optometry and Vision Science, School of Ophthalmology & Optometry and Eye Hospital, Wenzhou Medical University, Wenzhou 325027, China; 3Oujiang Laboratory (Zhejiang Laboratory for Regenerative Medicine, Vision and Brain Health), School of Ophthalmology & Optometry and Eye Hospital, Wenzhou Medical University, Wenzhou 325027, China

**Keywords:** alpha-synuclein, Parkinson’s disease, motor skill learning, fine movement, adenosine A_2A_ receptor

## Abstract

Parkinson’s disease (PD) is characterized pathologically by abnormal aggregation of alpha-synuclein (α-Syn) in the brain and clinically by fine movement deficits at the early stage, but the roles of α-Syn and associated neural circuits and neuromodulator bases in the development of fine movement deficits in PD are poorly understood, in part due to the lack of appropriate behavioral testing paradigms and PD models without motor confounding effects. Here, we coupled two unique behavioral paradigms with two PD models to reveal the following: (i) Focally injecting α-Syn fibrils into the dorsolateral striatum (DLS) and the transgenic expression of A53T-α-Syn in the dopaminergic neurons in the substantia nigra (SN, PITX3-IRES2-tTA/tetO-A53T mice) selectively impaired forelimb fine movements induced by the single-pellet reaching task. (ii) Injecting α-Syn fibers into the SN suppressed the coordination of cranial and forelimb fine movements induced by the sunflower seed opening test. (iii) Treatments with the adenosine A_2A_ receptor (A_2A_R) antagonist KW6002 reversed the impairment of forelimb and cranial fine movements induced by α-Syn aggregates in the SN. These findings established a causal role of α-Syn in the SNc-DLS dopaminergic pathway in the development of forelimb and cranial fine movement deficits and suggest a novel therapeutic strategy to improve fine movements in PD by A_2A_R antagonists.

## 1. Introduction

Parkinson’s disease (PD) is the second most common neurodegenerative disease, with an incidence of 1-2% of people over 60 years old [1,2]. Abnormal alpha-synuclein (α-Syn) accumulation and the progressive loss of dopaminergic neurons in the substantia nigra are the prominent pathological characteristics of PD patients. As a result of dopamine depletion in the substantia nigra pars compacta (SNc)-striatum system, PD patients present cardinal motor symptoms including resting tremor, rigidity, bradykinesia, and postural instability [3]. However, at the early phase, the most prominent symptom of PD patients is the deficits in skilled forelimb use. PD patients are presented with impairments in hand fine motor skills (such as pen holding, buttoning, and knotting) before the onset of cardinal motor dysfunctions. The fine movement impairments significantly affect daily routine activity (such as tying a shoe and wearing a shirt) and thus reduce the quality of life of PD patients [4]. The current clinical assessment of motor function, such as the Unified Parkinson’s Disease Rating Scale (UPDRS) [5], is not sufficiently sensitive for evaluating subtle deficits in fine movements in PD as executive cognition elements, such as motor skill learning, which are also involved in the control of fine movements [6]. At a PD clinic, fine movement skills are specifically assessed by neuropsychological test such as the finger tapping test (FTT) or the neurological test such as finger-to-thumb tapping. Current PD studies often focus on investigation into the impairments in gross motor function and coordination in PD models by evaluating locomotor activity, rotarod coordination, and gait function [7,8,9,10]. For example, some studies used the rotarod test to evaluate the motor coordination of rodents, which are particularly sensitive for detecting cerebellar dysfunction [11]. However, mice with striatal dopamine depletion show only mild or no motor deficits on the typical accelerating rotarod test [12]. Investigations into fine movement control are hampered by a lack of appropriate behavioral paradigms for the specific assessment of fine movement. To address this issue, we first employed the “sunflower seed opening” test in which mice had to use bilateral forelimbs and cranial motor activities to peel the shells of sunflower seeds to obtain the rewards within the required time [13]. Furthermore, we analyzed the deficits of the fine movement control of specific forelimbs by employing a “single pellet reaching” task in which mice had to use the over-trained, skilled, and preferred forelimb to reach for small food pellets, grasp them, and retrieve them toward the mouth for eating.

In addition to the behavioral paradigm limitation, there is still a lack of appropriate PD animal models for the specific assessment of fine movement control. The PD neurotoxin models by 6-OHDA and MPTP are often confounded by gross motor deficits, which can mask fine movement deficits. The transgenic PD mice with the overexpression of wild-type or PD-related mutant α-Syn only show scarce or low levels of α-Syn expression in midbrain dopaminergic neurons. To overcome these limitations, we leveraged the focal injection of α-Syn fibrils into a specific brain region to investigate the causal role of α-Syn aggregates in the nigrostriatal pathway during the development of fine movement deficits. Furthermore, we adopted a line of A53T α-Syn transgenic mice with the targeted expression of A53T α-Syn in the midbrain dopaminergic neurons under the control of the PITX3 promoter and conditionally regulated by the tet-off system [14] to determine critical control of fine movements by the nigro-striatal dopaminergic pathway.

Furthermore, the dopaminergic system plays an important role in integrating motor and cognitive processing considering that dopamine depletion and cortico-striatal pathway dysfunction contribute to cardinal gross motor symptoms and fine movement deficits in PD [15]. The adenosine A_2A_ receptors (A_2A_Rs) are expressed at high levels in the striatum to integrate the cortico-striatal glutamate signaling and nigra-striatal dopamine signaling and to control the cortico-striatal synaptic plasticity and cognition [16,17,18]. Striatal A_2A_Rs can also play a regulatory role in instrumental learning by co-localizing and forming heterodimers with the dopamine D2 receptors and metabotropic glutamate 5 receptors in the striatum [17]. The striatal A_2A_R activation exerts an inhibitory control of motor activity as well as various cognitive functions, including goal-directed behavior in normal mice [19,20] and action sequence learning under PD conditions [21]. Accordingly, we proposed that A_2A_R may represent a therapeutic target for alleviating fine movement deficits in PD.

In this study, we coupled two PD models to produce fine movement without gross movement deficits with two behavioral paradigms for the assessment of fine movement (i.e., “sunflower seed opening” and “single pellet reaching”) to establish the causal relationship between α-Syn abnormal aggregates in the nigro-striatal dopaminergic pathways and the impairments of fine movement. We first dissected the role of α-Syn aggregation in the nigro-striatal pathway in controlling fine movements by the focal injection of α-Syn fibrils into the dorsolateral striatum (DLS) and SNc. Furthermore, we identified the critical control of fine movement by α-Syn aggregating in the dopaminergic system by overexpressing α-Syn in midbrain dopaminergic neurons using PITX3-IRES2-tTA/tetO-A53T transgenic mice (A53T transgenic mice) [14]. Lastly, we investigated the ability of A_2A_R antagonists to reverse fine movement deficits induced by α-Syn aggregates.

## 2. Results

### 2.1. Coordination of Bilateral Forelimbs and Cranial Motor Function Was Selectively Impaired by α-Syn Abnormal Aggregation in SNc but Not in DLS

Since the DLS plays a critical role in motor skill learning, we first sought to identify whether the abnormal accumulation of α-Syn in DLS impaired coordination functions, including bilateral forelimbs and cranial motor. Quantitative analyses verified the damage effect caused by A53T α-Syn fibrils injections (Figure 1B,C, t_4_ = −5.019, *p* = 0.007). To exclude the gross motor confounding effects, we assessed the locomotion activity and found that there was no difference between the PBS and the α-Syn group (Figure 1D, t_20_ = 1.205, *p* = 0.242), demonstrating that α-Syn aggregations in DLS did not affect gross motor activity at 3 months after the injection. To further evaluate the coordination function, animals underwent the sunflower seed opening test (Figure 1E,F). In contrast, we did not find any differences between the two groups, because there was neither a main effect (F_(1,20)_ = 0.357, *p* = 0.557) nor a time × α-Syn injection interaction effect (F_(1.602,21.246)_ = 0.119, *p* = 0.749) (Figure 1G).

On the other hand, the nigro-striatal pathway was considered to be a significant neural circuit for reinforcement learning behaviors. We next injected α-Syn fibrils bilaterally into SNc (Figure 1H) to investigate the modulation effect of the nigro-striatal pathway on fine movement activity. We confirmed the Lewy body accumulation in SNc resembling the pathological characteristics of PD patients (Figure 1I). Quantitative analyses illustrated the remarkable damage effect (Figure 1J, t_4_ = −4.840, *p* = 0.008). This was confirmed by the notable decrease in TH-positive DA neurons in the SNc (Figure 1L). There was approximately a 30% DA depletion in the SNc of A53T α-Syn-injected mice compared to that with PBS-injected mice (Figure 1M, t_4_ = 4.021, *p* = 0.016). We then assessed the locomotion activity and revealed no significant differences between the PBS and α-Syn injection group (Figure 1K, t_27_ = 0.874, *p* = 0.390), excluding the confounding effect caused by gross motor functions. Importantly, in the sunflower seed opening test, we observed that mice with α-Syn injections obtained fewer seeds than the controls (Figure 1N, interaction effect of test duration × groups: F_(1.341,46.948)_ = 8.025, *p* = 0.003; between groups main effect: F_(1,35)_ = 7.851, *p* = 0.008), indicating that the coordination function by bilateral forelimbs and cranial nerves was significantly impaired by α-Syn aggregation in the SNc.

### 2.2. Skilled Forelimb Motor Activity Was Impaired by α-Syn Abnormal Aggregations in the DLS and SNc

Asymmetrically impaired dexterous skilled motor activity between the upper extremities is a prominent characteristic of PD patients, especially in the early phase of the disease. Hence, we investigated whether the abnormal aggregations of α-Syn in the DLS affected skilled forelimb movement. With the newly developed single-pellet reaching task (Figure 2A), mice were trained to use the preferred forelimb to reach the pellet, grasp it, retract the paw, and to bring the pellet back to its mouth; then, they were trained to consume it via a narrow slit. This test evaluates the comprehensive kinematics of dexterous skilled forelimb activity, including paw, joints, and muscles. Compared to mice injected with PBS, mice injected with A53T α-Syn fibrils in DLS showed a lower success rate, as evidenced by the significant difference between the group effect (F_(1,13)_ = 80.857, *p* = 0.0001) and testing sessions × groups interaction effect (Figure 2C, F_(1.720,22.362)_ = 0.087, *p* = 0.891). Moreover, the reaction time was notably reduced in the α-Syn injection group compared to the PBS group (Figure 2D, training sessions × treatment group: F_(1.966,25.553)_ = 0.024, *p* = 0.975; groups main effect: F_(1,13)_ = 64.750, *p* = 0.0001).

We further examined the effect of α-Syn abnormal aggregations in the SNc on skilled forelimb motor activity. As expected, success rates (Figure 2F, testing sessions × groups: F_(4,136)_ = 1.815, *p* = 0.129; between groups main effect: F_(1,34)_ = 27.550, *p* = 0.0001) and reaction times (Figure 2G, testing sessions × groups: F_(3.324,113.027)_ = 1.575, *p* = 0.195; between groups main effect: F_(1,34)_ = 8.240, *p* = 0.007) were dramatically impaired by SNc α-Syn abnormal aggregations. Thus, we speculate that the SNc-DLS pathway is the critical neural circuit for motor skill regulation.

### 2.3. Midbrain Dopamine-Specific Depletion by A53T Transgenic Mice Selectively Impaired Skilled Forelimb Motor Activity but Not the Coordination of Bilateral Forelimbs and Cranial Nerves

Although α-Syn fibrils injections into the DLS and SNc resulted in abnormal Lewy body aggregation in those regions and behavioral impairments, the specific role of midbrain dopaminergic neurons in the regulation of fine movements is still not well defined. To address this issue, we adopted a line of A53T transgenic mice with α-Syn specifically expressed in midbrain DA neurons, in which the A53T α-Syn expression was controlled by PITX3 gene and conditionally silenced by the tet-off system. This transgenic line was obtained by crossbreeding PITX3^+/IRES2-tTA^ heterozygous mice with the E2 line of tetO-A53T transgenic mice (Figure 3A). The DOX diet was withdrawn 2 months before the behavioral experiments (Figure 3B). The pathological characteristics of the transgenic line was confirmed by immunofluorescence staining (Figure 3C). Compared with control mice, A53T transgenic mice displayed prominent α-Syn accumulation (Figure 3D, t_4_ = −9.761, *p* = 0.001). A moderate decrease in DA neurons (approximately 30% decrease) was observed in the SNc of A53T transgenic mice by TH staining (Figure 3E,F, t_5.515_ = 2.811, *p* = 0.034) after DOX withdrawal for 2 months. The open field test did not detect any locomotion impairment caused by the transgenic line (Figure 3G, t_15.520_ = 0.131, *p* = 0.897). In the sunflower seeds opening test, the number of seeds obtained by A53T transgenic mice and controls increased over testing time indistinguishably (Figure 3H, between groups effect: F_(1,265)_ = 1.346, *p* = 0.254, testing time × groups interaction effect: F_(1,26.003)_ = 1.361, *p* = 0.254), implying the absent modulating effect on the coordination function of bilateral forelimbs and cranial nerves. Conversely, in the single-pellet reaching task, the success rates (Figure 3I, between groups effect: F_(1,21)_ = 19.108, *p* = 0.0001, testing sessions × groups interaction effect: F_(2.461,51.673)_ = 0.232) and the reaction time (Figure 3J, between groups effect: F_(1,21)_ = 14.696, *p* = 0.001, testing sessions × groups interaction effect: F_(4,84)_ = 0.310, *p* = 0.870) were readily impaired in A53T transgenic mice.

### 2.4. The Specific A_2A_R Antagonist KW6002 Recovered the Coordination Deficit Caused by SNc Dopamine Depletion Selectively

Currently, there is still a lack of an appropriate pharmacological therapeutic method to treat subtle motor deficit in PD patients. According to our recent studies [20,21,22] and others [23], activation striatal A_2A_Rs exert an inhibitory control of a variety of cognitive and motor functions, implying a potential role of A_2A_R antagonist in improving the motor skill function. Therefore, we investigated whether the specific A_2A_R antagonist KW6002 was able to recover the impaired coordination function and skilled forelimb motor activity. Interestingly, KW6002 did not improve skilled forelimb motor activity when α-Syn aggregation in SNc, as evidenced by the absence of effects on the single-pellet reaching task (Figure 4E, success rate, testing sessions × groups interaction effect: F_(4,80)_ = 0.037, *p* = 0.994; between groups effect: F_(1,20)_ = 0.510, *p* = 0.483 and Figure 4F, the reaction time, testing sessions × groups interaction effect: F_(4,80)_ = 0.173, *p* = 0.912; between groups effect: F_(1,20)_ = 0.320, *p* = 0.578). However, the pharmacological blockade of A_2A_Rs significantly increased the number of seeds obtained in the sunflower seed opening test (Figure 4D, testing time × groups interaction effect: F_(1,294,23.297)_ = 42.846, *p* = 0.001; between groups effect: F_(1,18)_ = 35.252, *p* = 0.0001). Therefore, the pharmacological blockade of A_2A_Rs selectively promoted the coordination function of bilateral forelimbs and cranial nerves, indicating its potential treating effect on the coordination deficit of PD patients.

## 3. Discussion

### 3.1. Alpha-Synuclein Aggregates in the SNc-DLS Pathway Distinctly Affects Forelimbs and Cranial Fine Movements

In addition to cardinal motor impairments, PD patients also experience disability with respect to impaired fine movement/manual dexterity with difficulty in routing tasks such as fastening buttons, tying shoelaces, and handwriting, significantly affecting their quality of life [24]. The impairments in fine movement differ from bradykinesia [25], which mainly includes impairments with respect to locomotion, velocity, and amplitude of movement. Skilled fine movement is acquired by practice over several sessions [26,27] and the successful execution of fine movement involves the control of gross movement as well as other motor coordination [28]. Recent studies have shown that the striatum and cortico-striatal interactions play an important role in controlling fine skill movement [26,29,30,31,32] and skill learning, which are evaluated by reach-to-grasp task paradigms [33]. The DLS is an important subcortical motor area that receives extensive monosynaptic inputs from both the M1 and M2 motor cortex [34,35,36]. The DLS also receives DA signal inputs from the SNc, which is crucial for fine movement regulations [37]. Accordingly, previous studies have shown that DLS inactivation or lesion impairs forelimb fine movements, but the grasping action was intact; i.e., animals were able to retrieve the pellet when the task was made easier by reducing the transport distance. This suggests that the DLS exerts an important effect on regulating the timing, amplitude, and the reliability of an action which is requisite for fine movement execution [13,28,32,38,39].

However, the neural circuit bases for fine movement under pathological (such as α-Syn aggregation) conditions are poorly defined. In the present study, we compared dexterous skilled forelimb motor activities and the coordination of bilateral forelimbs and cranial motor functions using two newly developed behavioral paradigms, i.e., single-pellet reaching task and sunflower seed opening test. The “single-pellet reaching” task mainly focuses on the use of a skilled forelimb to target the pellet, grasp it, and bring it to the mouth for consumption. On the other hand, the “sunflower seed opening” test involves varying degrees of both forelimbs and tongue/jaw function. This test relies heavily on the forelimbs to adjust the position of the seed so that they can bite off pieces of shells, thus representing a mixed task [13]. Our results revealed that α-Syn aggregates in DLS only impair skilled forelimb activity by the single-pellet reaching task, but it still preserves the motor coordination function of bilateral forelimbs and cranial motor nerves by the sunflower seed opening test and gross motor function by the locomotion test. Moreover, abnormal α-Syn aggregation in the SNc impairs both dexterous skilled forelimb use and the coordination function of bilateral forelimbs and cranial fine movements with intact locomotion activities. Thus, α-Syn aggregation in the nigro-striatal pathway impairs fine movement without the overt effect on ambulation in these PD models. Our findings of the impairment in fine movement by α-Syn aggregation in the DLS collaborate with previous reports that the nigro-striatal system influences forelimb behaviors by modulating the DLS’s neuron activities [13]; moreover, with the correlation of skilled fine and gross movements with a loss of forelimb movement representations in the motor cortex M1 and DLS [26], this observation confirms the critical role of the DLS in the development of fine movement deficits in PD. Moreover, the coordination function of bilateral forelimbs and cranial activity depends on more complicated neural mechanisms with a possibility of involving the dorsomedial striatum (DMS) and cerebellum [40,41].

With the different sensitivities of dexterous skilled forelimb motor activity and the coordination of bilateral forelimbs and cranial motor function relative to the α-Syn aggregation, the nigro-striatal and midbrain dopaminergic neurons are largely attributed to the gradual progression course of PD and the gradient functional heterogeneity of these pathways in the neural circuit [42]. The dexterous skilled forelimb motor activity was first affected due to α-Syn aggregation in the DLS. Progressively, as the entire SNc is affected, more complicated behavioral tasks, including the coordination control of both limbs and cranial motor function, become impaired, indicating that additional neural circuits such as “SNc-DMS”, “ventral tegmental area (VTA)-nucleus accumbens (NAc)”, or even “cerebellum-striatum” circuits are involved in regulating coordination behavior. Furthermore, other skill learning elements including instrumental learning [19] and sequence learning [21] may contribute to the DLS’s modulation of fine movement in PD.

### 3.2. Alpha-Synuclein in the Midbrain Dopaminergic Pathway Affects Forelimbs and Cranial Fine Movements

The nigro-striatal dopaminergic pathway plays a critical role in gross motor control, as evidenced by the cardinal motor symptoms and L-dopa-induced improvement of motor activity in PD. Although the motor symptoms of PD can be relieved by levodopa (L-dopa) treatments, they were unable to improve functional plasticity in the motor cortex [43] as well as motor skill learning [44]. Thus, it is important to critically evaluate the effect of dopamine depletion and α-Syn aggregation in the nigro-striatal pathway on a complex, finely coordinated and dexterous motor skill. Importantly, midbrain dopamine-specific depletion by A53T transgenic mice selectively impaired skilled forelimb motor activity but not the coordination of bilateral forelimbs and cranial nerves. This is consistent with recent studies demonstrating a role for dopamine in the control of motor skill learning [28,45,46]. The specific involvement of the SNc-DLS pathway in the control of skilled forelimb activity was confirmed by A53T transgenic mice. Before the onset of widespread neuronal loss in the transgenic line, there is a severe depletion with respect to dopamine release specifically in the DLS of 2-month-old A53T transgenic mice (i.e., the DOX-containing diet was removed after 2 months), the exact same time point for our experimental analysis [14]. On the other hand, the discrepancy between α-Syn fibrils injections into the SNc group and A53T transgenic line group in the sunflower seed opening test, i.e., the coordination function of bilateral forelimbs and cranial nerves, might partly be attributed to the specific impairment of dopaminergic neurons in A53T transgenic mice. Other neurons, e.g., glutamatergic and GABAergic neurons, and neurogliocyte were possibly involved when α-Syn fibrils were injected into SNc. The exact role of other cell types in the modulation of fine movement needs to be further investigated. The reversal of fine movements induced by α-Syn aggregations in the midbrain dopaminergic neurons can be attributed to two distinct functions of dopamine signaling, namely invigorating movement and improving motor skill learning by providing “reward prediction error” teaching signals. The “vigor” function of dopamine derives from the L-dopa responsiveness of bradykinesia in PD and is supported by extensive experimental evidence [47,48]. Furthermore, dopamine is required for the formation of the bidirectional plasticity of medium spiny neurons (MSNs), i.e., LTP in the D1 receptor expressing neurons and LTD in D2 receptor expressing neurons, providing the possible mechanism for motor skill learning [49]. The DA depletion resulted in the marked instability of synaptic connections and dysregulated synapses that are remodeled in the motor cortex. Pathologically, motor-learning-induced newly formed spines failed to stay stable and were eliminated immediately, possibly contributing to the impaired maintenance of motor leaning memory [50].

Current PD diagnosis in clinic is primarily performed on the clinical presentation of cardinal motor symptoms (with UPDRS) and L-dopa drug responses, although various molecular biomarkers and neuroimaging approaches have been attempted for early PD diagnosis. Our findings of the impairment of dexterous skilled forelimb activities with gross motor activities induced by α-Syn aggregation in the nigro-striatal dopaminergic pathway indicate a possible behavioral approach for assisting early PD diagnosis. The behavioral assessment in PD diagnosis has the unique advantages of being non-invasive, cost-effective, and easy to administer at clinic, home, or online by the participants themselves.

### 3.3. Pharmacological Blockade of A_2A_Rs more Likely Recovered the Coordination deficit of PD rather Than Skilled Forelimb Motor Activity

As dopamine replacement treatments markedly improve gross motor activity, their effects on motor skill learning and fine movement control are unclear; it is important to explore the non-dopaminergic therapeutic targets for the improvement of deficits in fine movement and motor skill learning in PD patients. Based on the selective colocalization of the A_2A_R and D_2_ dopamine receptor in the striatopallidal neurons and their antagonistic functional interaction in the striatum, the adenosine A_2A_R antagonist has been pursued for improving the motor symptoms of PD in recent decades; it has finally received approval in the US and Japan for the treatment of adult PD patients experiencing OFF time who are currently taking levodopa (plus a decarboxylase inhibitor) [51,52]. The overall effect size in reducing OFF time (−0.38 to −0.82 h) was comparable with other adjuncts relative to levodopa therapy—such as MAO-B inhibitors and COMT-inhibitors; however, the full potential of this drug class remains to be explored. Furthermore, our study [21] and other [23] studies demonstrate the ability of the A_2A_R antagonist to reverse impairments caused by α-Syn in motor skill learning, such as motor sequence learning. Therefore, we sought to investigate the exact effect of the blockade of A_2A_Rs on the improvement of subtle forelimb use and the more complicated coordination of both limbs and cranial motor function. Notably, the pharmacological blockade of A_2A_Rs selectively relieved the forelimbs and cranial motor function deficits caused by α-Syn aggregation in SNc. However, A_2A_R antagonist treatments did not improve the skilled forelimb motor activity deficits induced by α-Syn pathological accumulation in SNc. As the sunflower seed opening test may involve not only the forelimb fine movement but also more complicated coordination and executive control elements with the possible involvement of DMS, our results indicated that A_2A_Rs played a more significant role in coordination control, which requires a more demanding cognitive load. This is consistent with our recent study [19] showing that A_2A_R in DMS plays a predominant role in controlling model-based goal-directed behaviors, and this requires more analyses and computation [53]; moreover, we observed that the associative cortico-striatal (DMS) loop was the default model of striatal functions [54] and that a previous finding showed that the deletion of the indirect pathway in the DMS (but not DLS) produces pronounced psychomotor and cognitive effects [55].

The ability of A_2A_R antagonists to enhance fine movements may be attributed to A_2A_R acting as a cognitive “brake” controlling for a range of cognitive behaviors, including working memory [31], reversal learning [32], Pavlovian fear conditioning [33], set-shifting [34], goal-directed behavior [15,35], and motor sequence learning [17] in normal animals. On the other hand, the improvement effect on fine movement exerted by A_2A_R antagonists might be caused by the upregulated expression of A_2A_Rs in the hippocampus induced by α-Syn aggregation [56]. Furthermore, recent studies show that the A_2A_R blockade decreases α-Syn aggregation in Syn T-Synphilin-1 neuroglioma cells [57] and rescues synaptic and cognitive deficits in α-Syn-transgenic mice [58], and A_2A_R gene disruption protects in the α-Syn model of PD by preventing the loss of dopamine and dopaminergic neurons [59]. In contrast to targeting the hippocampus cortex by cholinesterase and NMDA inhibitors, the selective localization of A_2A_R in the striatopallidal pathway would provide a novel and promising target for selectively alleviating cognitive deficits in PD. This assumption was verified in our demonstration that A_2A_R antagonists reversed the action sequence learning deficit caused by A53T and WT α-Syn abnormal accumulation in SNc [21]. With the approval of the A_2A_R antagonist istradefylline [60] for treating PD patients by the FDA, the ability of A_2A_R antagonists, which not only enhanced gross motor activity but also improved cognition, led us to propose that A_2A_R antagonists may represent a novel therapeutic target for reversing fine movement deficits in PD patients (with the finger tapping or finger-to-thumb tapping test).

## 4. Materials and Methods

### 4.1. Animals

PITX3-IRES2-tTA/tetO-A53T transgenic mice were obtained by crossbreeding PITX3^+/IRES2-tTA^ heterozygous mice with *tetO*-A53T transgenic mice [14] (Figure 3A) (kindly provided by Peking University, Beijing, China). All animals were handled in accordance with the protocols approved by the Institutional Ethics Committee for Animal Use in Research and Education at Wenzhou Medical University, China. The transgenic mice used in the experiment included female and male mice about 12–14 weeks old, with a weight range of 19–27g. C57BL/6 male mice, at least 8 weeks old and 23–27g each, were used in the experiments. Mice were housed in an ambient temperature of 22 ± 0.5 °C and a relative humidity of 60 ± 2% with a 12h light/dark cycle. Mice were single-housed and underwent experiments in the light cycle. The numbers of animals used in all experiments were shown in the Appendix A.

### 4.2. Doxycycline Treatment

A regular diet was replaced with DOX-containing (200 mg/kg of diet) food pellets (Bio-Serv, Flemington, NJ, USA) to suppress the expression of A53T α-Syn from the early embryonic stages to the weaning age [14]. The DOX diet was then consistently supplied to the newly weaned mice until two months before the experiment, at which point the animals were back on a regular diet. Thus, α-Syn expression was controlled by the PITX3 gene and conditionally regulated by the tet-off system (Figure 3B).

### 4.3. Drug Administration

KW6002 ((E)-1,3-diethyl-8-(3,4-dimethoxystyryl)-7-methyl-3,7-dihydro-1H-purine-2,6-dioe), a selective adenosine A_2A_R antagonist, 5 mg/kg (Sundia, Walnut, CA, USA), was fully dissolved in a 1:1 mixture of dimethylsulfoxide (DMSO, Sigma, Burlington, MA, USA) and ethoxylated castor oil (Sigma). This mixture (30%) was then further diluted in water (70%) to obtain a KW6002 suspension. The control mice were treated with vehicle. Drugs were injected intraperitoneally (i.p.) routinely in a volume of 0.1 mL/10 g of body weight. In the single-pellet reaching task, drugs were administrated only during the testing phase (5 consecutive days) 30 min before testing. The drug was given every 6 h in the sunflower seed opening test.

### 4.4. Analysis of Recombinant α-Syn Protein and Stereotactic Surgery

The purification of recombinant α-Syn proteins and in vitro fibril formation was performed as previously described [61,62] and has been successfully used in our previous studies [21,56]. Briefly, the full-length cDNA of human α-Syn-His6 containing A53T mutations was synthesized and cloned into the *E. coli* expression vector pET24a. The expression vector was then transformed into BL21 (DE3) cells. An appropriate chromatography method, including nickel affinity and gel filtration, was implemented to purify the target proteins. α-Syn fibrils were briefly placed in an ultrasound bath before intracerebral injections. 

Conditional impairment was achieved by stereotaxic injections with α-Syn fibrils (the A53T locus mutation, 1.5 mg/mL) into the bilateral dorsolateral striatum (DLS, AP, +0.98 mm; ML, ±2.20 mm; DV, −2.60 mm) and SNc (AP, −3.16 mm; ML, ±1.25 mm; DV, −4.00 mm) 5 μg/side, and the control (PBS) was bilaterally injected into DLS and SNc with a speed of 0.25ug/min and the retention of the needle for 10 min after the injection for full diffusion. Based on our pilot studies, the time required for establishing the animal model was 1 month for SNc and 3 months for DLS [61].

### 4.5. Locomotion

To evaluate general locomotor activities, mice were tested in an open field arena two days before the behavioral test. The locomotion test was conducted by employing the standard open field apparatus (40 cm × 40 cm × 35 cm) with a video camcorder fixed at the top. On the first day, each mouse was placed in an open field cage to habituate for 5 min. On the second day, each mouse was recorded individually for 15 min. The total distance traveled in 15 min was analyzed by the EthoVision XT program (Noldus, Wageningen, The Netherlands).

### 4.6. Single-Pellet Reaching Task

The single-pellet reaching test was performed as previously described [63], and the mice were trained to reach through an opening to retrieve food pellets (45 mg) placed on an elevated grid floor that was indented 2 cm away from the front wall (Figure 2A). Skilled forelimb use and motor learning were evaluated by a custom-made clear Plexiglas chamber (8 cm × 15 cm × 20 cm) with 3 vertical slits (0.5 cm × 13 cm) made in the front wall. A stage (0.9 cm tall) was placed in front of the slits to hold the food pellets. Mice were food-restricted to keep 85–90% of their initial body weight throughout the experiment. Animals underwent the training/shaping phase (for 3–5 days) before the testing phase (5 consecutive days). Both the training/shaping session and testing phase were terminated until the mouse spent 20 min in the chamber. Mice that showed greater than 70% preference for either hand (more than 70% reaches are performed with one forelimb) were used for the test. During the testing process, the mice had to stretch out the preferred forelimb via the slit to reach the pellet and then grasp it; the mice then retracted the paw and brought the pellet back to its mouth and consumed it. “Reach” was scored each time an animal extended its forelimb through the slits. A “Successful” attempt was scored if the animal smoothly grasped the food and placed it in its mouth. “Success rate” was calculated by “the Successful attempts/Reach”. “Reaction time” was defined by “the Successful attempts/Time”. Finally, quantitative statistics were made depending on the “Success rate” and “Reaction time” in the testing phase.

### 4.7. Sunflower Seed Opening Test

To test object manipulation abilities, each mouse was placed individually into a clear plastic arena with 30 sunflower seeds located on one side. Mice were allowed to have free access to sunflower seeds for 24 h. The mice needed to pick the sunflower seeds up with their forepaws and obtain the seed’s kernel by peeling the shells of the seeds. The primary outcome measure for sunflower seed testing was the total number of peeled seeds at 1 h, 3 h, 8 h, and 24 h.

### 4.8. Immunohistochemistry

On the second day after all behavioral experiments were performed, mice were deeply anesthetized with avertin (Sigma) and then transcardially perfused with 0.01 M PBS (pH = 7.4) followed by an ice-cold 4% paraformaldehyde wash. Immunohistochemistry was performed on 30 μm free-floating sections. The free-floating sections were washed in 0.01 M PBS (pH = 7.4) and then incubated for 60 min in PBS containing 0.3% Triton X-100 and 15% normal donkey serum. Primary antibodies were incubated following manufacturer protocols: tyrosine hydroxylase (abcam; polyclonal antibody; ab112; 1:1000), anti-alpha-synuclein (phospho S129) (Wako; monoclonal antibody; 015-25191; 1:1000), and anti-alpha-synuclein (abcam; monoclonal antibody; ab138501; 1:300). An Alexa 568-conjugated secondary antibody (Invitrogen; 1:1000) was used to visualize the staining. For immunohistochemical analysis, the sections were immunostained using the avidin-biotin complex (ABC) system (VectastainABC Elite Kit; Vector Laboratories, Burlingame, CA, USA), and immunocomplexes were visualized with chromogen 3.3′-diaminobenzidine. Sections were counterstained with hematoxylin and images were acquired with a brightfield microscope. Fluorescence images were captured using a laser-scanning confocal microscope (Leica DM6B). TH-positive neurons and α-Syn quantitative data were obtained by counting the number of TH-positive neurons and α-Syn-expressed neurons in the non-overlapping field of the intact unilateral SNc region under a 20-fold microscope. P-Syn was calculated by counting the number of neurons that contain Phosphorylated α-Syn in a single 20-fold field of vision in DLS and SNc. About seven to nine brain slices from each animal were counted, and their averages were taken. Three to five animals in each group were included in the quantitative staining analyses.

### 4.9. Statistical Analysis

All data were presented as mean ± SEM and were processed with SPSS 20.0. Independent sample *t*-tests were used for immunofluorescence staining, and locomotion data analyses. Two-way ANOVA for repeated measurements were used with testing sessions as the within-subject effect and different α-Syn fibrils as between-subject effects, with *p* < 0.05 as representing a statistically significant result.

## Figures and Tables

**Figure 1 ijms-24-01365-f001:**
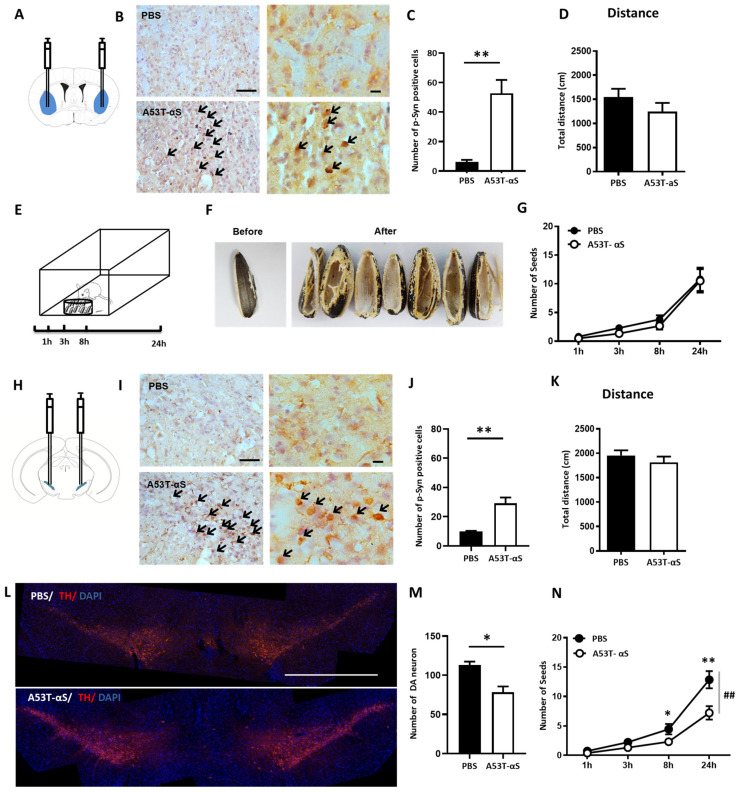
The coordination function was selectively impaired by abnormal α-Syn aggregation in SNc but not in DLS. (**A**) Sketch map of α-Syn bilateral injection into DLS. (**B**,**C**) Immunohistochemistry stain with phosphorylated α-Syn illustrating that A53T α-Syn fibrils obviously induced Lewy body inclusions (black arrows head) in DLS (scale bars: left, 50 μm; right, 10 μm), and quantitative analyses confirmed the damaging effect led by A53T α-Syn (t_4_ = −5.019, *p* = 0.007, PBS: n = 3, A53T: n = 3, independent samples *t*-test). (**D**) Locomotor activity was not affected by α-Syn pathological accumulation (t_20_ = 1.205, *p* = 0.242, PBS: n = 10, A53T: n = 12, independent samples *t*-test). (**E**) Schematic design of the sunflower seed opening test. Each mouse was placed individually into a clear plastic arena with 30 sunflower seeds located on one side. The total number of seeds obtained by mice was recorded at the end of 1, 3, 8, and 24 h. (**F**) The presentation of sunflower seeds obtained by mice. (**G**) There was no significant difference in the sunflower seed opening test between PBS and A53T α-Syn groups (testing time × group interaction effect: F_(1.602,21.246)_ = 0.119, *p* = 0.749, PBS: n = 10, A53T: n = 12, two-way ANOVA for repeated measurements). (**H**) Sketch map of A53T α-Syn bilateral injection into SNc. (**I**) Representative immunohistochemical images of Lewy body (black arrows head) aggregation in SNc (scale bars: left, 50 μm; right, 10 μm). (**J**) Quantitative analyses confirmed the damaging effect led by A53T α-Syn (t_4_ = −4.840, *p* = 0.008, PBS: n = 3, A53T: n = 3, independent samples *t*-test). (**K**) Mice with or without A53T α-Syn injections into the SNc did not show any difference in the locomotion test (t_27_ = 0.874, *p* = 0.390, PBS: n = 18, A53T: n = 11, independent samples *t*-test). (**L**) Immunofluorescent images of TH positive neurons in the coronal sections of PBS and A53T α-Syn injection into SNc mice (scale bars: 1 mm). (**M**) Number of TH-positive neurons in the SNc (t_4_ = 4.021, *p* = 0.016, PBS: n = 3, A53T: n = 3, independent samples *t*-test). (**N**) The coordination function was significantly impaired by α-Syn aggregation in the sunflower opening test (testing time × group interaction effect: F_(4,136)_ = 1.815, *p* = 0.143; between groups effect: F_(1,34)_ = 27.550, *p* = 0.0001, PBS: n = 18, A53T: n = 19, two-way ANOVA for repeated measurements; * *p* < 0.05, ** *p* < 0.01, and ^##^ *p* < 0.01).

**Figure 2 ijms-24-01365-f002:**
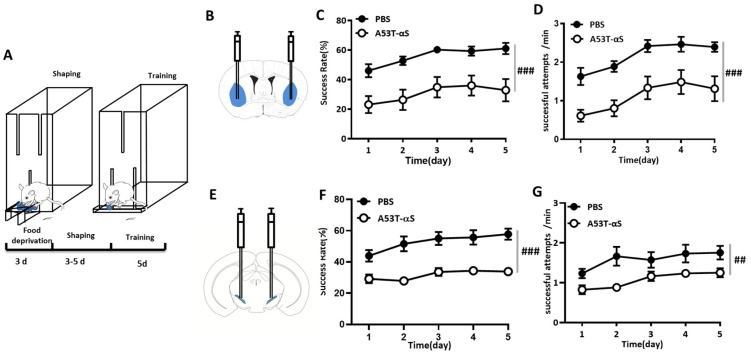
Skilled forelimb motor activity was impaired by α-Syn abnormal aggregations in the DLS and SNc. (**A**) Schematic design of the single-pellet reaching task paradigm. (**B**) Sketch map of α-Syn bilateral injection into DLS. (**C**) The success rate was obviously impaired by A53T α-Syn aggregation in DLS in the single-pellet reaching task (testing sessions group interaction effect: F_(1.720,22.362)_ = 0.087, *p* = 0.891, between groups effect: F_(1,13)_ = 80.857, *p* = 0.0001, PBS: n = 8, A53T: n = 7, two-way ANOVA for repeated measurements). (**D**) The reaction time was also significantly reduced in mice with A53T α-Syn injection in DLS (testing sessions x group interaction effect: F_(1.966,25.553.362)_ = 0.024, *p* = 0.975; between groups effect: F_(1,13)_ = 64.750, *p* = 0.0001, PBS: n = 8, A53T: n = 7, two-way ANOVA for repeated measurements). (**E**) Sketch map of A53T α-Syn bilateral injection into SNc. (**F**,**G**) A53T α-Syn aggregation in SNc produced prominent impairing effect on success rates (testing sessions × group interaction effect: F_(4,136)_ = 1.815, *p* = 0.129; between groups effect: F_(1,34)_ = 27.550, *p* = 0.0001, PBS: n = 17, A53T: n = 19, two-way ANOVA for repeated measurements) and reaction time (testing sessions × group interaction effect: F_(1.341,46.948)_ = 8.025, *p* = 0.003; between groups effect: F_(1,35)_ = 7.851, *p* = 0.008, PBS: n = 17, A53T: n = 19, two-way ANOVA for repeated measurements; ^##^ *p* < 0.01, ^###^ *p* < 0.001).

**Figure 3 ijms-24-01365-f003:**
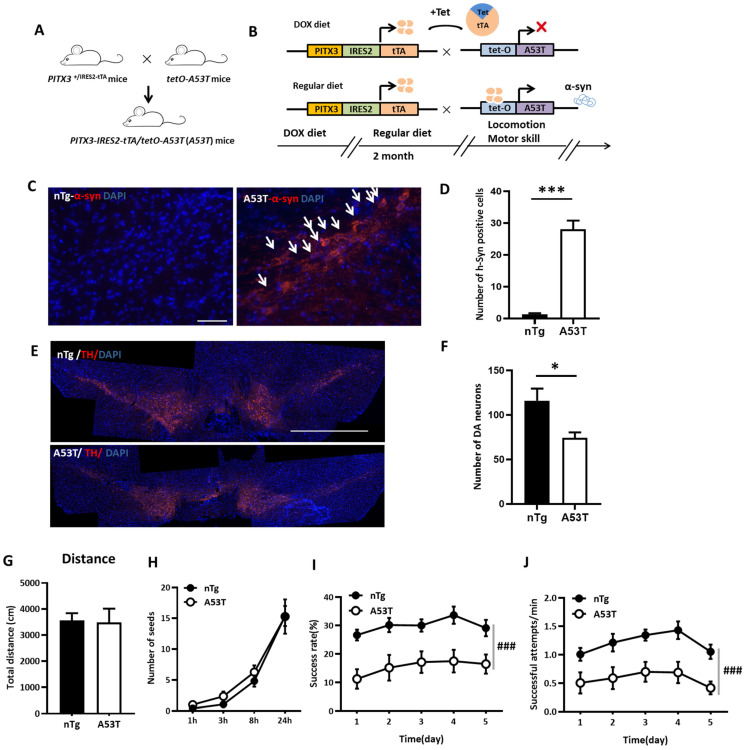
Midbrain dopamine-specific depletion by A53T transgenic mice selectively impaired skilled forelimb motor activity but not the coordination of bilateral forelimbs and cranial nerves. (**A**) The diagram depicts the generation of PITX3-IRES2-tTA/tetO-A53T (A53T transgenic mice) transgenic mice by crossbreeding PITX3-IRES2-tTA and tetO-A53T mice. (**B**) The tet-off system: top: to suppress the expression of A53T α-Syn, a DOX-containing diet was provided before behavioral experiments. Down: A53T α-Syn was induced in order to be expressed after DOX-containing diet withdrawal. (**C**) Immunofluorescent images show the selective expression of A53T α-Syn (white arrows head) in SNc after DOX diet withdrawn 2 months (scale bars: 50 μm). (**D**) The number of Lewy body inclusions in A53T transgenic mice compared to that of nTg mice (controls) (t_4_ = −9.761, *p* = 0.001, nTg: n = 3, A53T: n = 3, independent samples *t*-test). (**E**) The midbrain DA neurons were visualized by TH immunofluorescent staining in the SNc of A53T transgenic mice and the control nTg mice (scale bars: 1 mm). (**F**) Quantitative analysis showed that the number of TH-positive neurons was reduced by approximately 30% in A53T transgenic mice compared to that of nTg mice (t_5.515_ = 2.811, *p* = 0.034, nTg: n = 5, A53T: n = 5, independent samples *t*-test). (**G**) There was no statistical difference in the locomotion test between the A53T transgenic mice and the nTg mice (t_15.520_ = 0.131, *p* = 0.897, nTg: n = 19, A53T: n = 11, independent samples *t*-test). (**H**) All mice performed identically in sunflower seed opening test (testing time × group interaction effect: F_(1,26.003)_ = 1.361, *p* = 0.254, nTg: n = 17, A53T: n = 11, two-way ANOVA for repeated measurements); between groups effect (F_(1,265)_ = 1.346, *p* = 0.254, nTg: n = 17, A53T: n = 11, two-way ANOVA for repeated measurements). (**I**,**J**) The success rates (testing sessions × group interaction effect: F_( 2.461,51.673)_ = 0.232, *p* = 0.837; between groups effect: F_(1,21)_ = 19.108, *p* = 0.0001, nTg: n = 14, A53T: n = 9, two-way ANOVA for repeated measurements) and reaction time (testing sessions × group interaction effect: F_(4,84)_ = 0.310, *p* = 0.870; groups main effect: F_(1,21)_ = 14.696, *p* = 0.001; * *p* < 0.05, *** *p* < 0.001, and ^###^
*p* < 0.001) were notably reduced in A53T transgenic mice in the single-pellet reaching task.

**Figure 4 ijms-24-01365-f004:**
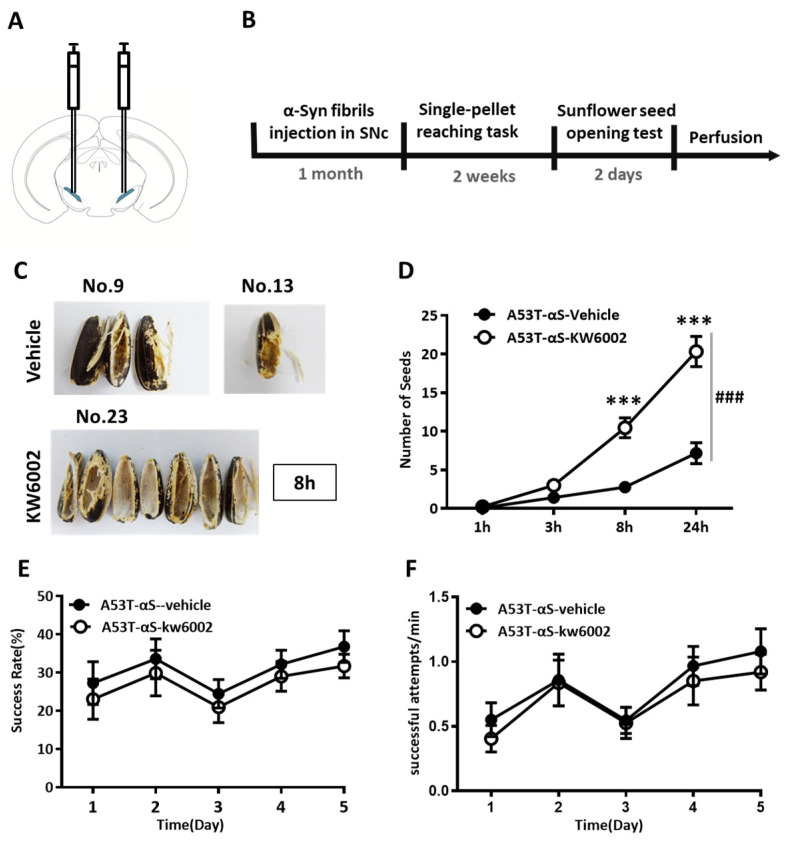
The specific A_2A_R antagonist KW6002 recovered the coordination deficit caused by SNc dopamine depletion selectively. (**A**) Sketch map of A53T α-Syn bilateral injections into SNc. (**B**) The time course of the α-Syn fibers injection and behavioral experiments. (**C**) The representative sunflower seed shells peeled by A53T α-Syn injected in the SNc mice with KW6002 treatments versus the vehicle at the end of 8 h of the sunflower seed opening test. (**D**) Mice treated with KW6002 showed significant improvement of the coordination function in the sunflower seed opening test (testing time × group interaction effect: F_(1,294,23.297)_ = 42.846, *p* = 0.001; between groups effect: F_(1,18)_ = 35.252, *p* = 0.0001, A53T-αS-Vehicle: n = 12, A53T-αS-KW6002: n = 9, two-way ANOVA for repeated measurements). (**E**,**F**) The success rates (testing sessions × group interaction effect: F_(4,80)_ = 0.037, *p* = 0.994; between groups effect: F_(1,20)_ = 0.510, *p* = 0.483, A53T-αS-Vehicle: n = 12, A53T-αS-KW6002: n = 10, two-way ANOVA for repeated measurements) and reaction time (testing sessions x group interaction effect: F_(4,80)_ = 0.173, *p* = 0.912; between groups effect: F_(1,20)_ = 0.320, *p* = 0.578, A53T-αS-Vehicle: n = 12, A53T-αS-KW6002: n = 10, two-way ANOVA for repeated measurements; *** *p* < 0.001, and ^###^ *p* < 0.001) of SNc A53T α-Syn aggregation mice were not recovered by KW6002, as evidenced by the single-pellet reaching task.

## Data Availability

Data are available upon reasonable request from the corresponding author.

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
