# Peer review of "α-Synuclein Aggregates in the Nigro-Striatal Dopaminergic Pathway Impair Fine Movement: Partial Reversal by the Adenosine A2A Receptor Antagonist"

_ijms, 2023, doi:10.3390/ijms24021365_

Round 1

Reviewer 1 Report

The early pathogenic stages of Parkinson's disease are poorly characterized. This study uses two experimental models of PD (stereotactic bilateral injections of A53T alpha synuclein fibrils into DLS and SNc; and A53T transgenic mice) to detect changes in fine motor movements using reaching task to assess skilled forelimb motor activity and sunflower seed opening test to assess coordinated bilateral forelimbs and cranial motor functions. The authors also used a well known A2AR antagonist KW6002 to try to reverse any of the fine motor movement deficits induced by alpha-syn aggregation. While the results are of interest to the readers, there are several major inconsistencies in the available data and lack of methodological details in certain sections that must be addressed, as enumerated below:

1. Regarding the sunflower opening test, there appears to be a model-dependent effect on the number of seeds obtained by animals. That is, the i.p. injected A53T alpha-syn fibrils appeared to decrease the number of sunflower seeds opened (Fig. 1N, from approx. 13 to 7, PBS vs A53T), whereas in the transgenic A53T mice there appears to be no effect on cranial and forelimb fine movements (Fig. 3H, aprrox. 15 seeds for transgenic and non-transgenic groups).  This occurs despite the fact that the authors reported similar levels of reduction of dopaminergic neurons in SNc (approx.. 30% in i.p. injected alpha-syn A53T (Fig. 1M) vs. approx. 40% in transgenic animals (Fig. 3F)). The authors need to address this discrepancy in the Discussion. Additional immunohistochemistry/confocal imaging data may be required to provide a mechanistic basis for this difference in response. For example, with i.p. injection manipulations, are there certain cell types that are altered/activated by the alpha-syn fibrils that are not induced in the transgenic animals which could affect the sunflower opening test behaviour?

2. Important negative controls for Fig. 4B (e.g., non-transgenic vehicle, non-transgenic + KW6002) seem warranted to increase the validity of the apparent effect of KW6002 on improving coordination of forelimb and cranial motor functions.  It is puzzling why the A53T-alpha-syn vehicle control animals have such low seed number (Fig 4B, approx. 7 seeds opened at 24 h) compared to the A53T-alpha-syn (no vehicle) animals shown in Fig. 3H (showing approx. 15 seeds opened at 24 h). It would seem necessary to determine whether the observed increase in number of seeds opened at 8 and 24 h (approx. 10 and 20, respectively) in the A53T-KW6002 animal group can be similarly seen in non-transgenic animals treated with KW6002. Since KW6002 was injected i.p. every 6 h, could this acute stressful manipulation result in the suppression of fine motor movement? A brief discussion of this possibility may be included in the Discussion.  Also, the authors need to indicate the background of non-transgenic mouse controls used (is it just C57BL/6 mice or others).

3. Also as the authors already acknowledged, the striatum is a brain region with high density of A2ARs. What is the basis for the KW6002-induced improvement in fine motor movement? Is it due to greater suppression of plasma membrane expressed A2ARs? Does A53T alpha-syn cause an increase in A2AR surface expression? What is the effect of KW6002 on locomotor response (need to add this data in the open field graph in Figu3G). Does KW6002 reduce anxiety levels in open field test (need to present % center square crossings). Since A2ARs may be able to form complexes with different species of alpha-syn at the plasma membrane and thereby potentially increasing neurotoxicity/neurodegeneration, do the authors have data showing that the A53T alpha-syn mutants increase levels of A2ARs in both SNc and DLS? This potential further upregulation of A2ARs might explain the higher levels of neuronal loss in SNc and higher number of sunflower seeds opened when KW6002 was administered (20 seeds in Fig. 4B vs. 15 seeds in Fig. 3H).   

4. The cell types that take up the A53T alpha-syn fibrils in DLS (Fig. 1B) and SNc (Fig. 1I) need to be identified.  For example, do TH-positive neurons colocalize with phosphorylated alpha-syn in Fig. 1L? Are there non-neuronal cells that are activated and take up alpha-syn fibrils in DLS or SNc? Furthermore, does i.p. injection of A53T alpha-syn in DLS lead to retrograde transport of alpha-syn fibrils in SNc, and does i.p. injection ofA53T alpha-syn in SNc lead to migration of alpha-syn fibrils in DLS? Why does it take 3 months to establish the alpha-syn fibril PD model when fibrils are injected in DLS vs. 1 month if injected directly into SNc (does it take this long for alpha-syn fibrils from DLS to migrate to SNc  in order to see 30-40% reduction of SNc neurons?).

5. Missing information in Fig legends or Methods section include the number of animals used (no n-values indicated anywhere), brief description of Open Field Test, the source of alpha-syn fibrils used in this study not mentioned.

6. There are numerous typo/grammatical errors to list here.  A thorough spell check/proofreading is recommended. Also, Fig. 3D (typo h-syn instead of p-syn).

Minor:

1. A brief mention of the possible heteroprotomeric complexes between A2ARs and other adenosine receptor subtypes and between A2ARs and dopaminergic receptors in the striatum in the Introduction might be needed.

Reviewer 2 Report

In the current study, the authors investigated the role of α-Syn in the dopaminergic SNc-DLS pathway involved in forelimb and cranial fine movement deficits. Furthermore, they suggest a new therapeutic strategy to improve fine movements in PD by A2AR antagonists.

Some revisions need to be done before acceptance.

1)  General Revision:

-       Typography: the authors should read thoroughly their manuscript and check: 1) space between words; 2) English of some sentences

-       Line 370: please, correct “Therecore” with “Therefore”.

2)  Material and Methods section:

-       In the animal section, please, specify sex, age and weight of transgenic mice.

-       Please, add some recent references in all materials and methods section.

-       Please, justify the use of KW-6002 suspended in dimethylsulfoxide, ethoxylated castor oil and water with a proportion of 15%:15%:70%. Moreover, I suggest specifying the period of KW-6002 treatment.

-       Specify the time of sacrifice. In addition, specify the type of anesthetic used to perform sacrifice.

-       Please, specify the code, commercial brand, city and country of provenance for all materials.

3)  Results:

-       For a better understanding of the text, we recommend adding the type of mouse on which the analysis was performed.  

-       Please, enhance the resolution of all images and in all immunofluorescences images add arrows to indicate positive cells.

Round 2

Reviewer 1 Report

The authors have made additional efforts to address all my major concerns, although there are still significant typo/grammar errors throughout the manuscript (see below) which the authors should quickly address.

Minor corrections for authors:

Introduction

p. 1, line 28, “casual role of α-Syn” change to “causal role of α-Syn”

p. 1, line36, “…incidence of 1-2 ‰” change to “…incidence of 1-2 %”

p. 1, line 40, “compacta (SNc)-“ change to “pars compacta (SNc)-“

p. 2, line 68, “6-OHDA and MTPT” change to “6-OHDA and MPTP”

p. 2, line 74, “expression A53T α-Syn” change to “expression of A53T α-Syn”

p. 2, line 84, “…by co-localizing with and form…” change to “…by co-localizing with and forming…”

p. 2, line 86, “The striatal A2AR exert” change to “The striatal A2AR exerts”

Results

p. 5, line 133, “α-Syn abmormal aggregation” change to “α-Syn abnormal aggregation”

p. 5, line 137, “(D) Locomition” change to “(D) Locomotion”

p. 6, line 160-1, “to reach the pellet, grasp it, retracted the paw and brought the pellet back to its mouth, then consumed it” change to “to reach the pellet, grasp it, retract the paw and bring the pellet back to its mouth, then to consume it”

p. 6, line, line 165, “significant between groups effect” change to “significant difference between groups effect”

p. 8, line 196, “with α-Syn specifical expression” change to “with α-Syn specifically expressed”

p. 8, line 205, “after DOX withdraw” change to “after DOX withdrawal”

Fig. 3 legend, change “withdraw” to “withdrawal”, “reduced by approximately 30% in A53T”, “There was no statistical difference”

Discussion

p. 12, line 272, “SNc-DLS pathway distinctly affect” change to “SNc-DLS pathway distinctly affects”

p. 13, line 314, define “DMS”, “is largely attribute to the gradually progression” change to “is largely attributed to the gradual progression”

p. 14, Line 354, “…when α-Syn fibrils injection into SNc.” change to “ when α-Syn fibrils were injected into SNc.”

p. 14, line 355, “…need to be further investigate.” Change to “…need to be further investigated.”

p. 14, line 363, “DA depletion resulted in remarked instability” change to “DA depletion resulted in marked instability”, “synapsis” to “synapses”, “impaired of” change to “impairment of”, “dopaminergic pathway indicates”, “approach for assisting”, “advantages of being non-invasive”

p. 14, line 378, “markedly improves gross motor”, line 383 “A2AR antagonists has been pursued”

p. 14, line 368, “Current PD diagnosis in clinic primarily on clinical presentation…” change to “Current PD diagnosis in clinic is primarily made on clinical presentation…”

p. 15, line 402, “goal-directed behavior which needs more analyses”, line 433 “Peking University”

p. 15, line 411, “…A2AR antagonists might caused” change to “…A2AR antagonists might be caused”, “these pathways neural” change to “these pathways of neural”

p. 15, line 413-414, change “…rescue synaptic and cognitive deficits in a-Syn-transgenic mice [61] and A2AR gene disruption” to “…rescues synaptic and cognitive deficits in α-Syn-transgenic mice [61], and that A2AR gene disruption”

p. 18, line 505, “Frist” to “First”, line 513 “Free-floating sections”, line 518 “was used to visualize”
